# Hormone Receptor Expression in Multicentric/Multifocal versus Unifocal Breast Cancer: Especially the VDR Determines the Outcome Related to Focality

**DOI:** 10.3390/ijms20225740

**Published:** 2019-11-15

**Authors:** Alaleh Zati zehni, Sven-Niclas Jacob, Jan-Niclas Mumm, Helene Hildegard Heidegger, Nina Ditsch, Sven Mahner, Udo Jeschke, Theresa Vilsmaier

**Affiliations:** 1Department of Obstetrics and Gynecology, Ludwig Maximilian University of Munich (LMU), Maistraße 11, 80337 Munich, GermanyHelene.Heidegger@med.uni-muenchen.de (H.H.H.); sven.mahner@med.uni-muenchen.de (S.M.); Theresa.Vilsmaier@med.uni-muenchen.de (T.V.); 2Department of General, Visceral, Transplant, Vascular and Thoracic Surgery, LMU, Marchioninistraße 15, 81377 Munich, Germany; Sven.jacob@med.uni-muenchen.de; 3Department of Urology, LMU Munich, University Hospital, Marchioninistraße 15, 81377 Munich, Germany; janniclas.mumm@med.uni-muenchen.de; 4Department of Obstetrics and Gynecology, University Hospital Augsburg, Stenglinstr. 2, 86156 Augsburg, Germany; nina.ditsch@uk-augsburg.de

**Keywords:** breast cancer, multifocal, unifocal, hormone receptor, estrogen receptor, progesterone receptor, vitamin D receptor, triple negative

## Abstract

The aim of this study was to evaluate the prognostic impact that hormone receptor (HR) expressions have on the two different breast cancer (BC) entities—multifocal versus unifocal BC. As the prognosis determining aspects, we investigated the overall survival (OS) and disease-free survival (DFS) by univariate and multivariate analysis. To underline the study’s conclusions, we additionally considered the histopathological grading and the tumor node metastasis (TNM) staging. A retrospective analysis was performed on survival-related events in a series of 320 breast cancer patients treated at the Department of Gynecology and Obstetrics at the Ludwig Maximillian University in Munich between 2000 and 2002. All three steroid receptors analyzed by immunohistochemistry, namely, the estrogen receptor (ER), the progesterone receptor (PR), and the vitamin D receptor (VDR), showed a significantly positive influence on the course of the disease, but only for the unifocal breast tumor patients. The prognosis of patients with multifocal breast cancer was either not affected by estrogen and/or progesterone receptor expression or even involved a worse etiopathology for the vitamin D receptor-positive patients. The estrogen receptor in unifocal breast cancer and the vitamin D receptor in multifocal breast cancer were especially identified as an independent prognostic marker for overall survival, when adjusted for age, grading, and staging. Altogether, our results strengthen the need to further investigate the behavior of the hormone receptors in breast cancer and understand why they have different effects on each focality type. Moreover, the studies for an adopted vitamin D supplementation due to breast cancer focality type must be enlarged to fully comprehend the remarkable and interesting role played by the vitamin D receptor.

## 1. Introduction

According to the global burden of disease study published in 2018, the most frequent cancer among the female population is, and has been for decades, breast cancer (BC). It is the leading cause not only of cancer deaths but also of disability-adjusted life-years (DALYs) among women worldwide, with 1.7 million incident cases, 535,000 deaths, and 14.9 million DALYs in 2016 [1].

So far, BC is still the focus of great attention, where much research has already been done on the disease. In the last decade in particular, work on the endocrine component has led to a major improvement in the endocrine therapy regimes for BC, used in adjuvant, neoadjuvant, and metastatic settings.

Endocrine therapy regimes have decreased the BC-associated mortality rate by approximately 30% and the risk for relapse by 40%, which is significant regardless of the age. Thus, these regimes are currently, according to National Comprehensive Cancer Network (NCCN) guidelines, an indispensable part of the adjuvant therapy for women with hormone receptor-positive (HR+) BC [2,3]. Clinical studies have confirmed that a high HR expression, such as estrogen receptor (ER), progesterone receptor (PR), and vitamin D receptor (VDR), increases the course of the disease [4,5,6,7,8]. In this study, we decided to further investigate the above-mentioned HR expressions, all being nuclear receptors, activated by their steroid hormones and being important targets for both prevention and treatment of BC. 

The VDR is expressed in healthy as well as in cancer cells. Activated by its hormonal ligand calcitriol (1,25-dihydroxyvitamin-D), it exerts its actions in an endocrine, paracrine, and autocrine manner. When calcitriol binds to the VDR, it causes a dimerization with the retinoid X receptor (RXR). The calcitriol–VDR–RXR complex binds to vitamin D response elements (VDREs) in various regulatory regions, which induce transcriptional regulation of gene expression. These target genes intervene in molecular pathways, resulting in the diverse vitamin D-mediated anticancer effects—inhibiting proliferation, inflammation, angiogenesis, invasion, and metastasis and, on the other hand, stimulating apoptosis and differentiation [9]. In this context, especially the two calcitriol-metabolizing catabolic enzymes CYP24A1 and CYP27B1 in cancer tissue gained importance for the carcinogenic effects of the pathway [10,11]. Compelling evidence implies that the usual balance between CYP24A1 and CYP27B1 becomes dysregulated during carcinogenesis, leading to the abrogation of the tumor suppressive effects triggered by VDR [8].

VDR expression fluctuates in the mammary gland during the maturation of the female body, initiating during puberty and peaking during pregnancy and lactation [8,12]. In BC, VDR expression is inversely associated with a higher cancer incidence, disease progression, and worse prognosis [5,13]. In any case, only the ER, PR, and HER2 status are currently considered as important predictive and prognostic immunohistochemical markers in the therapy decision-making [4,5,14]. For the prophylaxis of the cancer therapy’s side effects, a vitamin D supplementation with a dose between 800 and 1000 IU/d is common [15]. Already in 2009, Goodwin et al. showed for the first time that there was a significant relation between low vitamin D (25-hydroxyvitamin D) serum levels and a higher risk for relapse and mortality for BC patients [16,17,18]. Furthermore, compelling evidence was provided for the tumor suppressor function of VDR and agonists in breast tissue [9,19]. 

The prognostic roles of the HR and HER2 status were finally incorporated into the main factors for the 8th edition of the American Joint Committee on Cancer (AJCC) Cancer Staging Manual, determining prognostic stage groups as well as TNM classification and histologic grade [20]. BC is defined as HR+ if ≥1% of the tumor cells are immunohistochemically stained either ER- or PR-positive, which is the case in 75−80% of all diagnosed BC. As HR+BC is responsible for most of the deaths from the disease, estrogen and progesterone deprivation continues to be a mainstay of treatment for all stages of HR+ breast tumors. While the HR+/HER2- subtype involves the best prognosis, triple-negative (TN) BC, meaning HR-/HER2-, has unfavorable clinicopathological features and the poorest prognosis of all BC subtypes [21]. So far, studies that point out the benefit of a high VDR as well as HR expression under the aspect of the BC focality type are still lacking.

Since there is no clear standard international definition of the terms multifocal and multicentric [2,22] and for a distinct comparison with the unifocal patients, we merged the multifocal and multicentric BC patients into one multifocal group in this paper. Nonetheless, the most established understanding of multifocal BC is a single tumor with two or more foci and of multicentric BC as multiple primary tumors within the same breast [23,24]. Generally, unifocal BC is associated with a better prognosis, including OS and DFS, than multifocal and/or multicentric BC with identical tumor size. Weissenbacher et al. [25] even declare the focality as an independent prognostic factor and hypothesize that it should be considered in the current TNM classification of the UICC. In a subsequent study they demonstrated that the downregulation of E-cadherin multicentric/multifocal compared to unifocal BC with identical TNM staging might be connected with the worse prognosis of this tumor type [26]. In contrast, Fushimi et al. [27] did not find the focality to be predictive of a worse prognosis in their study of 734 BC patients since it did not significantly influence the OS. Nevertheless, they confirmed that multifocal and multicentric BC is associated with a worse DSF (*p* = 0.004). Another retrospective study, including 507 patients, claimed similar DSF and OS rates for each focality type and the lymph node status to be the only statistical significant factor affecting the prognosis [28]. In 2010, Tot et al. analyzed the HR expression (ER, PR, HER2) in 875 cases of multifocal and unifocal BC, where they could not verify significant differences between these two tumortypes regarding HR expression [29].

Being aware of the infinite studies about BC, several HRs, and the immense progress of today’s clinical oncology toward personalized treatments, we focused on the leading question of how HR expression influences the prognosis of each focality type. The aim of the study was to give a scientific basis for future BC endocrine therapy adjusted to the focality type, which may allow a even more effective and less toxic treatment in the future.

## 2. Results

### 2.1. Estrogen Receptor

#### 2.1.1. Unifocal BC

The expression of ER showed a statistically significant difference in the OS, DFS, histopathological grading, and staging by TNM classification for the unifocal BC patients.

The Kaplan–Meier curve revealed that an ER expression is statistically correlated with a better OS, which was additionally supported by the log-rank test with *p* = 0.005 for the unifocal group (Figure 1). Considering the histopathological tumor grading by the WHO and the TNM staging of the unifocal BC patients, the analysis showed that the unifocal BC patients benefit from being ER-positive (Table 1). Furthermore, ER-positive, unifocal BC patients were more frequently graded at a lower tumor stage (G1/2) than a high-grade tumor (G3/4) (boxplot visualization and Kruskal–Wallis calculation of *p* = 0.029). Additionally, this patient group had a significant lower risk for the presence of metastasis, also shown through boxplots and Kruskal–Wallis (*p* = 0.020). Neither the primary tumor size (pT *p* = 0.267) nor the lymph node status (pN *p* = 0.736) were influenced by ER expression. Supporting these results, Cox regression revealed the ER expression to be an independent prognostic marker for the OS (HR 0.282, 95% CI 0.114–0.698, *p* = 0.006) in this patient cohort (Table 2).

#### 2.1.2. Multifocal BC

The multifocal group revealed no significant difference between ER-positive or -negative patients regarding the OS (*p* = 0.238), DFS (*p* = 0.052), histopathological grading (*p* = 0.262), and all three categories of the TNM classification (pT *p* = 0.590, pN *p* = 0.430, pM *p* = 0.433) (Table 1).

### 2.2. Progesterone Receptor

#### 2.2.1. Unifocal BC

Evaluating the data, the PR expression also showed having a positive impact for the BC patients being affected by the unifocal type. Again, the same statistical devices were used to clarify and interpret the collected data. The Kaplan–Meier curve showed that unifocal BC patients have a better overall survival when being PR-positive, which was confirmed by the log-rank test with a *p*-value of 0.012 (Figure 2). Not only the OS of the unifocal BC patients but also the histopathological grading and the TNM staging were influenced positively by the expression of PR (Table 3). Boxplots showed that unifocal BC patients are more often graded having a well-differentiated tumor (G1) and being staged without metastasis (pM0), which was statistically confirmed with significant *p*-values, calculated with Kruskal–Wallis (*p* = 0.007 for the grading, *p* = 0.008 for the pM). The DFS (*p* = 0.070), tumor size pT (*p* = 0.390), and involution of local lymph nodes (*p* = 0.662) of Group 1 was calculated by Kruskal–Wallis to be insignificantly influenced by PR presence. Contrary to the ER, the PR was identified as a dependent prognostic factor, when conducting Cox regression (HR 0.453, 95% CI 0.161–1.279, *p* = 0.135) in the unifocal group (Table 4).

#### 2.2.2. Multifocal BC

Like the ER, the multifocal group was not significantly affected by the PR in none of the analyzed items, namely, OS *p* = 0.090, DFS *p* = 0.063, grading *p* = 0.087, pT *p* = 0.830, pN *p* = 0.313, pM *p* = 0.484 (Table 3).

### 2.3. Triple-Negative Breast Cancer

#### 2.3.1. Unifocal BC

The results of the receptors mentioned above matched the results we obtained analyzing the TN cases, which showed that unifocal TN BC, that is, ER, PR, and Her2new receptor-negative, has a significant worse OS, illustrated by Kaplan–Meier curves and the *p*-value of 0.05 calculated by log-rank tests (Figure 3).

However, no significant difference was observed for this group regarding the DFS (*p* = 0.952), histopathological grading (*p* = 0.317), and the TNM classification (pT *p* = 0.656, pN *p* = 0.756, pM *p* = 0.302), calculated with boxplots and Kruskal–Wallis tests (Table 5).

#### 2.3.2. Multifocal BC

Statistical analysis validated no significant different outcome regarding the multifocal TN group regarding the OS *p* = 0.556, histopathological grading *p* = 0.247, and the TNM classification (pT *p* = 0.874, pN *p* = 0.130, pM *p* = 0.98). This patient cohort showed with a *p*-value of 0.019 to have a worse DFS.

### 2.4. Vitamin D Receptor

#### 2.4.1. Unifocal BC

Statistically significant positive correlations were also proven for the unifocal BC patients, when expressing the VDR (Table 6). Unlike the ER and PR, the VDR influences neither the OS (*p* = 0.627) nor the DSF (*p* = 0.647) significantly (Figure 4a,b) but is statistically correlated with a better outcome regarding the histopathological grading and TNM staging for the unifocal group. Unifocal BC patients with a VDR expression were tested to be more often graded G1 with a well-differentiated tumor (*p* = 0.020), also staged according to TNM more frequently pT1 (*p* = 0.035), which means a primary tumor size of 2 cm or less and pN0 (*p* = 0.032), meaning there are no cancer cells in any nearby lymph nodes. In any case, Cox regression revealed the VDR expression to be a dependent prognostic marker for the OS (HR 1.086, 95% CI 0.114–0.698, *p* = 0.327) (Table 7).

#### 2.4.2. Multifocal BC

Another difference between ER, PR, and the VDR is that a VDR expression had a significant negative effect on the multifocal BC patients. VDR-positive, multifocal patients have a worse DFS (*p* = 0.000) than VDR-negative patients (Figure 4d). The OS in this patient group was not significantly (*p* = 0.106) influenced by the presence of the VDR (Figure 4). More detailed analysis of the intensity and distribution of VDR expression according to the IRS score (1–4) revealed that only a VDR expression equal to or higher than an IRS of 3 influences the DFS significantly (*p* = 0.024). Looking at the TNM classification, patients are more often staged pM1 having metastasis (*p* = 0.003) (Table 6). OS (*p* = 0.229), grading (*p* = 0.195), pT (*p* = 0.343), and pN (*p* = 0.313) showed not to be significantly affected by the presence of the VDR. Multivariate analyses were conducted, after adjusting for age, grading, and staging, and showed that the expression of VDR is an independent prognostic factor for the OS in the multifocal BC group (HR 0.804, 95% CI 0.921–1.280, *p* = 0.019) (Table 8).

## 3. Discussion

The aim of this study was to evaluate the prognostic impact that HR expression has on the two different BC entities—multifocal versus unifocal BC. All three analyzed steroid receptors showed a significantly positive influence on the course of the disease, but only for the unifocal breast tumor patients. The prognosis of multifocal BC was either not affected by ER and/or PR expression or even involved a worse etiopathology for the VDR-positive patients.

In regard to the focality, a consistent number of studies have demonstrated a strong correlation between multifocal BC and a higher risk of nodal involvement compared to women with unifocal BC [30,31,32]. While some authors found the focality to be an independent prognostic factor by multivariate analysis, others claim that only the sum of the invasive diameters of multifocal BC could have an important impact on the prognosis [25,27]. As we did not evaluate the data regarding the quality of the single aspect of focality, our study cannot contribute to the discussion. Nevertheless, no study to date has examined the prognostic impact that HRs have on the single tumor entities.

We are currently moving toward the era of personalized treatment, which is why knowledge regarding TNM stage and histopathological grading alone, although it supplies independent prognostic information, is inadequate for optimum patient treatment. Consequently, molecular biomarkers, such as ER and PR status, can provide prognostic information and additionally predict the response to therapy, and they have therefore become the center of research [7,33,34,35,36,37,38,39,40,41,42].

Nicolini et al. observed the identification of the ER status to be of rather predictive use, as it indicates the response to endocrine therapy, rather than a prognostic factor [34]. Furthermore, it was found that the presence of PR in ER-positive patients provides an additional independent predictive value because those with ER-positive and PR-positive BC were more likely to respond to endocrine therapy than women with ER-positive and PR-negative BC, which is consistent with several other study outcomes [35,36,43]. In contrast, our study evaluated by multivariate analysis that the presence of PR provided no independent predictive value for the OS. Nonetheless, we recommend the continuation of the measurement of both ER and PR in all newly diagnosed cases of BC, since it seems to have a positive impact on the prognosis for unifocal BC, although as a dependent biomarker.

In a recently published study of a small patient cohort, containing 21 matched pairs, Müller et al. observed that a high ERβ expression significantly correlates with poor OS and DFS only in patients with unifocal BC [23]. Comparing these results to our data, we also found that the prognostic value of the ER differs according to tumor focality and exclusively shows its impact on unifocal BC. Interestingly (considering that we merely analyzed the ERα), an ERβ expression shows a completely contrary impact on the prognosis of BC, which implies the further need to investigate these two receptor subtypes. It would also be of crucial relevance to understand why the ER, regardless of the subtype ERα or ERβ, and the PR do not influence the prognosis of multifocal BC at all.

Vitamin D and the VDR have become a complex story with an evolving landscape, especially in the last few decades [4,5,9,12,18,19,44,45,46,47,48]. Deuster et al. reviewed the importance of Vitamin D and its receptor in the pathogenesis of certain types of cancer such as colorectal, lung but mainly gynecological cancers, where they found the VDR to be upregulated, indicating its relevance for cancer etiology [46]. Al-Azhri et al. found in their large study of 1114 patients that a low VDR expression in BC was associated with a higher pT, ER, and PR negativity and TN BC expression. No associations could be drawn between a VDR expression and lymph node involvement or patient survival outcomes [47]. A growing body of epidemiologic evidence suggests that a deficient VDR expression is associated with a more aggressive disease, which has led to the current standardized vitamin D supplementation for BC prevention and therapy [15,47,49]. In contrast, our study has shown with a highly significant *p*-value of 0.00 that a VDR expression in multifocal BC leads to a worse DFS. Therefore, we suggest a critical reconsideration of vitamin D as a target subjected to downregulation along the BC progression and continued further research of the VDR pathogenesis in BC subtypes.

## 4. Materials and Methods

### 4.1. Patients

The basis for all the research conducted was a patient cohort with BC treated at the Department of Gynecology and Obstetrics at the Ludwig Maximillian University in Munich in the years 2000 to 2002.

As the study’s aim was to find differences in the course of the disease between unifocal and multifocal BC in relation to the HR status, it was necessary to set HRs. We decided to include, and further investigate by immunohistochemistry (Figure 5), the following HRs: ER, PR, VDR, and the TN cases.

For the determination of the focality, patients were required to undergo set clinical diagnostics, namely, clinical examination, ultrasonic, and X-ray. If the focality identification was still not possible, diagnostics had to be added such as nuclear magnetic resonance imaging (NMRI), pneumocystography, or galactography. Patients with an unclear HR status and/or focality were excluded from the database used for the study.

A total collective (TC) of 320 patients all meeting the requirements mentioned above remained (Table 9). Considering the focality, the TC was subdivided into two groups—one group with unifocal BC patients for a total of 173 (Group 1) and the second group made up of 147 multifocal BC patients, also containing the 21 multicentric as well as multicentric and multifocal BC patients (Group 2).

For each group, the OS and DFS analysis was performed and afterwards compared, always regarding the HR status. To underline the study’s conclusions, we decided to additionally consider the differences in the histopathological grading [50] and TNM staging [51] between the groups.

Detailed patient characteristics from the TC are displayed in Table 10. Total patient numbers (*n*) in the subgroups differ because we could not obtain all information from each patient regarding each subgroup. The median patient age of our 320 consecutive patients included in the total was 58 years (range 69). Of these, 54.0% were diagnosed with unifocal and 45.9% with multifocal and/or multicentric BC. Of the patients, 49.8% had histological carcinoma of no special type (NST). The majority of our cohort had a low-grade tumor (G1 or G2 70.9%) and a tumor size smaller than 2 cm (pT1, 67.1%; pT2, 28.4%; pT3, 1%; pT4, 3.4%). Finally, 57.5% of all patients were staged pN0.

### 4.2. Immunohistochemistry

Samples selected for the immunohistochemistry to identify the HR status were processed according to the published [5,52] and well-established methods described in brief below. Figure 5 shows a selection of the immunohistochemical stained HRs. Primary antibodies used for the staining were anti-ER alpha (monoclonal rabbit IgG) for the ER and anti-VDR (mouse IgG2a) for the VDR. Negative as well as unstained tissue appeared blue in contrast to the positive-stained cells, which appeared in a brownish color (Figure 5).

Therefore, a combination of pressure-cooker heating and the standard streptavidin–biotin–peroxidase complex with the mouse IgG-Vectastain Elite ABC kit (Vector Laboratories, Burlingame, CA, USA) was used. Antibodies used for the staining were anti-ER, anti-PR, anti-VDR, and anti-Her2new for the TN cases.

First, the paraffin-fixed tissue sections were dewaxed with xylol (15 min) and rehydrated in three ascending series of alcohol (70–100%). For epitope retrieval, sections were subjected for 10 min in a pressure cooker using sodium citrate buffer with a pH of 6.0, containing 0.1M sodium citrate in distilled water and 0.1M citric acid. After washing in phosphate buffered saline (PBS), endogenous peroxidase activity was quenched by immersion in 3% hydrogen peroxide (Merck; Darmstadt, Germany) in methanol for a duration of 20 min. Non-specific binding of the primary antibodies was prevented by incubating the sections with diluted normal serum (10 mL PBS containing 150 µl horse serum; Vector Laboratories) at room temperature for 20 min. Then, the 5 µm thick sections were incubated for one hour at room temperature with the primary antibodies. Repeated PBS washing steps and incubation with the avidin–biotin–peroxidase complex (diluted in 10 mL PBS; Vector Laboratories) for 30 min followed. Visualization was achieved with substrate and chromogen 3,3’-diaminobenzidine (DAB; Dako, Glostrup, Denmark) for 8–10 min.

Finally, sections were counterstained with Mayer‘s acidic hematoxylin, dehydrated in an ascending series of alcohol (50–98%), treated with xylol, and covered [5,52].

To analyze the intensity and distribution pattern of antigen expression, we used the semi-quantitative immune-reactive score of Remmele and Stegner [53]. The staining intensity (Score 0 = no staining, Score 1 = weak staining, Score 2 = moderate staining, Score 3 = strong staining) needed to be multiplied with the percentage of positively stained cells (Score 0 = 0% stained cells, Score 1 ≤ 10% stained cells, Score 2 = 10–50% stained cells, Score 3 = 51–80% stained cells, Score 4 ≤ 80% stained cells) to get the IR Score, which reached from 0 to 12. We assessed a score from 0–1 to be receptor-negative, and every score ≥2 to be receptor-positive. The staining intensity of the slides was estimated optically by two blinded observers. Smaller sections, for determining the percentage of positively stained cells, were examined with a Leitz microscope (Wetzlar, Germany) and a 3CCD color camera (JVC, Victor Company of Japan, yokohama, Japan).

### 4.3. Statistical Analysis

Statistical analysis was accomplished using the computer software “Statistical Package for the Social Sciences” (IBM SPSS Statistics, version 24.0, IBM Inc., Chicago, IL, USA).

Differences between the initially defined receptors (always relating to the focality) influencing the OS and DFS were tested for significance. Using the TC database, Kaplan–Meier curve analysis was performed for each HR and, by applying the chi-square of the log-rank test, the significance was determined. In order to analyze the TNM staging and the WHO grading, and due to the separation based on focality, we created two groups indicated as Group 1 and Group 2. Boxplots and Kruskal–Wallis were used for statistical analysis. In order to evaluate whether the HRs were an independent prognostic factor, multivariate analyses via Cox regression were conducted. Each statistical test in our study was required to have a *p*-value less than 0.05 to be considered significant.

### 4.4. Ethics approval and consent to participate:

The tissue samples used in this study were leftover material after all diagnostics had been completed and were retrieved from the archives of Gynecology and Obstetrics, Ludwig Maximilian University, Munich, Germany. All patients gave their consent to participate in the study. All patient data were fully anonymized, the study was performed according to the standards set in the Declaration of Helsinki 1975. The current study was approved by the Ethics Committee of the Ludwig Maximilian University Munich, Germany (approval number 048–08, 18 March 2008). Authors were blinded from the clinical information during experimental analysis.

## 5. Conclusions

In conclusion, the present study concluded that the prognostic value of HR expression differs according to tumor focality and significantly correlates with a good prognosis but only in patients with unifocal BC. In patients with multifocal BC, the HR expression (ER and/or PR) did not prove a prognostic value for those cases, as it did not influence any of the four set points. The outcome of the TN cases underlines the fact that a lack of HRs is associated with a worse prognosis and, again, only for patients with unifocal BC. Nonetheless, the only independent prognostic marker, among all those investigated receptors regarding the OS in the unifocal group, was the ER. The VDR expression showed that it played a remarkably paradox role for BC prognosis. For unifocal BC patients, VDR expression proves to have significant positive correlations regarding grading and TNM staging. This is contrary to the multifocal BC patients, who have a worse DSF and are more often staged pM ≥1, when expressing the VDR. Furthermore, multivariate VDR analysis in the multifocal group even identified the VDR to be an independent prognostic marker. These data strengthen the need to further investigate the behavior of the VDR in BC and enlarge the studies for an adopted vitamin D supplementation due to the BC focality type. To the best of our knowledge, this work is the first study that investigates the influence that HRs and especially VDR and its expression on the different focality types of BC. The representativeness and importance of our study are provided by the large TC of 320 patients and the highly significant results we obtained by analyzing the data. These findings may contribute to novel progress and provide a promising perspective for innovation in the endocrine therapy regimes for premenopausal women with BC.

## Figures and Tables

**Figure 1 ijms-20-05740-f001:**
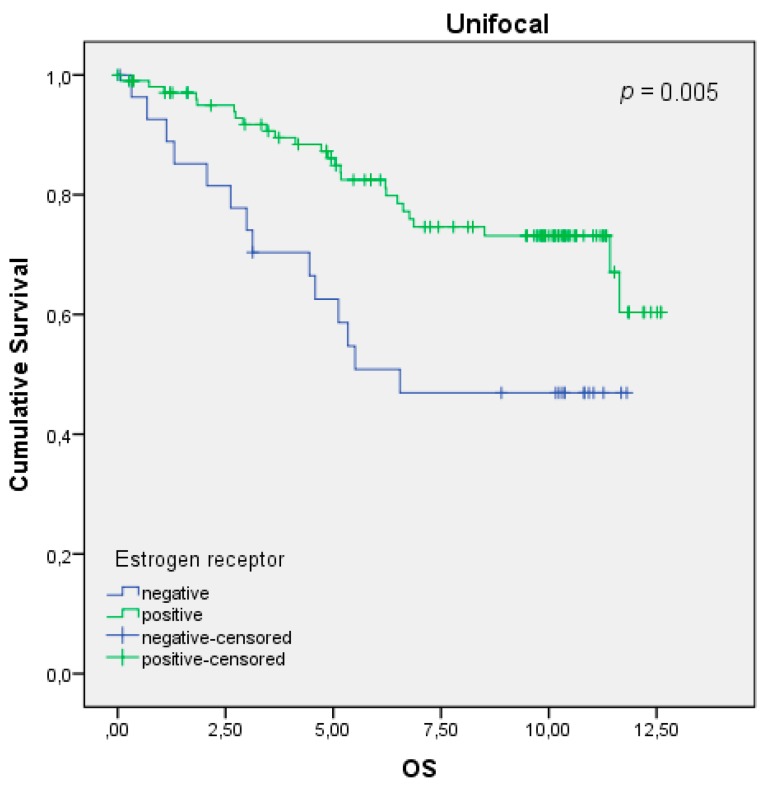
Kaplan–Meier survival analysis among estrogen receptor (ER)-positive and -negative patients. Overall survival (OS) of patients with unifocal breast cancer (BC) in years.

**Figure 2 ijms-20-05740-f002:**
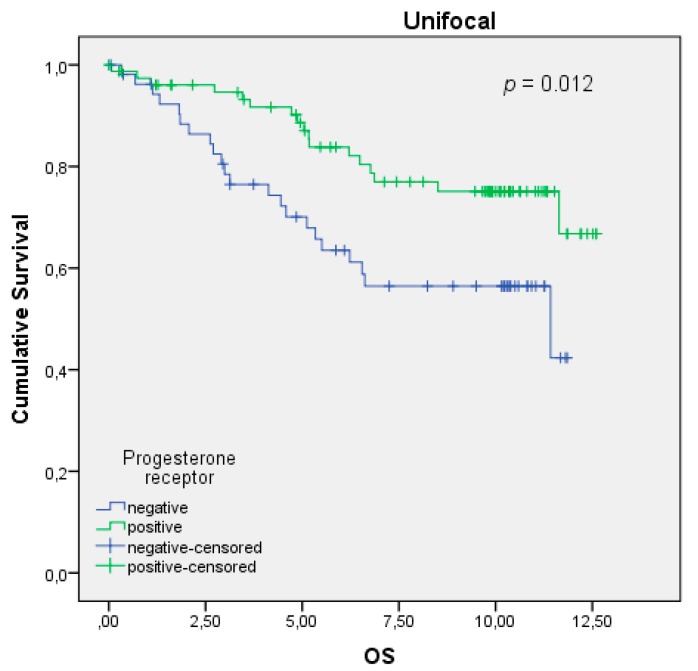
Kaplan–Meier survival analysis among PR-positive and -negative patients. OS of patients with unifocal BC in years.

**Figure 3 ijms-20-05740-f003:**
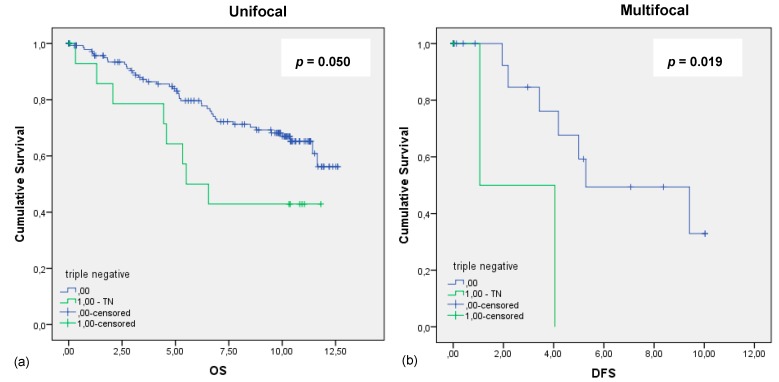
Kaplan–Meier survival analysis of the TN and HR-positive patients. (**a**) OS of patients with unifocal BC in years; (**b**) DFS of patients with multifocal BC in years.

**Figure 4 ijms-20-05740-f004:**
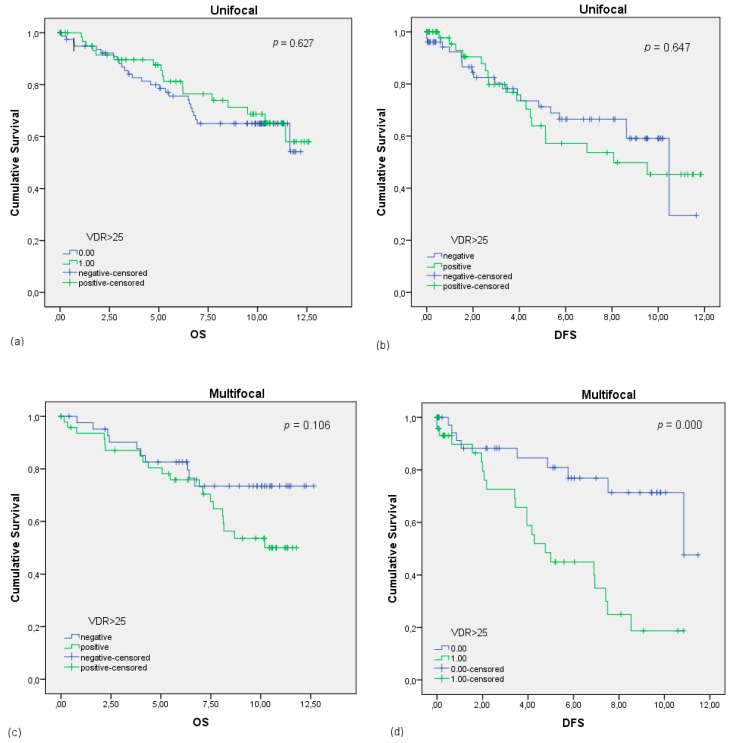
Kaplan–Meier survival analysis in VDR-negative and -positive patients. **(a)** OS of patients with unifocal BC in years. (**b**) DFS of patients with unifocal BC in years. (**c**) OS of patients with multifocal BC in years. (**d**) DFS of patients with multifocal BC in years. The VDR is defined to be positive (green curve) with an Immunoreactive score (IRS) ≤ 25.

**Figure 5 ijms-20-05740-f005:**
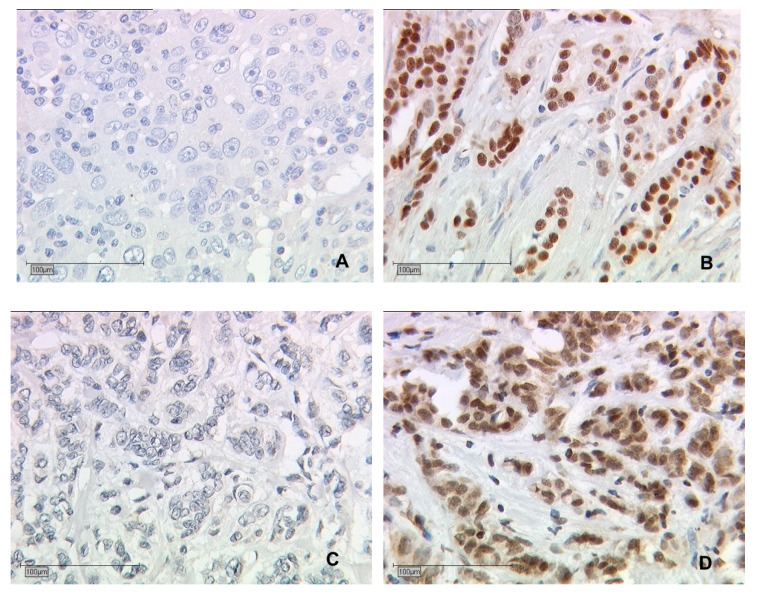
Immunohistochemical staining of ER and VDR after incubation with the primary antibody of the malignant breast cancer cells (25× lens). (**A**,**B**) Immunohistochemical staining of ER in human BC. (**A**) An IRS ≤1 meaning ER-negative and (**B**) an IRS value ≥2 being ER-positive. (**C**,**D**) Immunohistochemical staining of VDR in BC. (**C**) An IRS ≤1 meaning VDR-negative and (**D**) an IRS value ≥2 being VDR-positive.

**Table 1 ijms-20-05740-t001:** Significant results for the estrogen receptor-positive patients.

Estrogene Receptor	Unifocal	Multifocal
Overall survival	**• +**	
Disease-free survival		
Grading	**• +**	
pT		
pN		
pM	**• +**	

**•** = Expression of the particular receptor has a significant influence on the marked characteristics. **+** indicates Receptor expression affects the marked characteristics significantly positively.

**Table 2 ijms-20-05740-t002:** Multivariate Cox regression analysis of unifocal patients regarding OS.

Variable	Coefficient	HR (95% CI)	*p*-Value
Age	0.059	1.061 (1.023–1.100)	0.002
Grading	0.790	2.204 (0.987–4.923)	0.054
pT	−0.234	0.791 (0.560–1.118)	0.185
pN	0.552	1.737 (1.302–2.318)	0.000
pM	3.025	20.591 (5.681–74.640)	0.000
ER	−1.267	0.282 (0.114–0.698)	0.006

HR, hazard ratio; CI, confidence interval.

**Table 3 ijms-20-05740-t003:** Significant results for the progesterone receptor-positive patients.

Progesterone Receptor	Unifocal	Multifocal
Overall survival	**•** **+**	
Disease-free survival		
Grading	**• +**	
pT		
pN		
pM	**•** **+**	

**•** = Expression of the particular receptor has a significant influence on the marked characteristics; **+** indicates Receptor expression affects the marked characteristics significantly positively.

**Table 4 ijms-20-05740-t004:** Multivariate Cox regression analysis of unifocal patients regarding OS.

Variable	Coefficient	HR (95% CI)	*p*-Value
Age	0.051	1.052 (1.015–1.091)	0.006
Grading	0.537	1.711 (0.720–4.066)	0.224
pT	−0.129	0.879 (0.624–1.237)	0.459
pN	0.418	1.519 (1.175–1.965)	0.001
pM	2.670	14.443 (4.203–49.634)	0.000
PR	−0.791	0.453 (0.161–1.279)	0.135

HR, hazard ratio; CI, confidence interval.

**Table 5 ijms-20-05740-t005:** Significant results for the triple-negative patients.

Triple-Negative	Unifocal	Multifocal
Overall survival	**•** **-**	
Disease-free survival		**•** **-**
Grading		
pT		
pN		
pM		

**•** = Expression of the particular receptor has a significant influence on the marked characteristics; **-** indicates Receptor expression affects the marked characteristics significantly negatively.

**Table 6 ijms-20-05740-t006:** Significant results for the vitamin D receptor-positive patients.

VDR	Unifocal	Multifocal
Overall survival		
Disease-free survival		**•** **-**
Grading	**•** **+**	
pT	**•** **+**	
pN	**•** **+**	
pM		**•** **-**

**•** = Expression of the particular receptor has a significant influence on the marked characteristics; **+** indicates Receptor expression affects the marked characteristics significantly positively; **-** indicates Receptor expression affects the marked characteristics significantly negatively.

**Table 7 ijms-20-05740-t007:** Multivariate Cox regression analysis of unifocal BC patients regarding OS.

Variable	Coefficient	HR (95% CI)	*p*-Value
Age	0.073	1.075 (1.036–1.116)	0.000
Grading	0.643	1.903 (0.901–4.020)	0.092
pT	−0.056	0.945 (0.674–1.325)	0.744
pN	0.296	1.345 (1.070–1.690)	0.011
pM	3.212	24.839 (7.439–82.940)	0.000
VDR	0.082	1.086 (0.921–1.280)	0.327

HR, hazard ratio; CI, confidence interval.

**Table 8 ijms-20-05740-t008:** Multivariate Cox regression analysis of multifocal BC patients regarding OS.

Variable	Coefficient	HR (95% CI)	*p*-Value
Age	0.050	1.052 (1.022–1.083)	**0.001**
Grading	−0.001	0.999 (0.992–1.007)	0.864
pT	0.642	1.900 (1.394–2.589)	**0.000**
pN	0.011	1.011 (1.000–1.022)	**0.043**
pM	2.390	10.914 (4.456–26.731)	**0.000**
VDR	−0.218	0.804 (0.671–0.964)	**0.019**

Significant results are shown in bold; HR, hazard ratio; CI, confidence interval.

**Table 9 ijms-20-05740-t009:** Patient characteristics of the total collective, sample size of receptor expression.

Receptor Expression	Unifocal N_Total_ (%)	Multifocal N_Total_ (%)
ER+	105 (42.6)	96 (39.0)
ER-	28 (11.4)	17 (7.0)
PR+	79 (32.1)	75 (30.4)
PR-	54 (21.9)	38 (15.4)
VDR+	66 (24.3)	60 (22.0)
VDR-	89 (32.7)	57 (20.9)
TN	15 (75.0)	5 (25.0)

**Table 10 ijms-20-05740-t010:** Patient characteristics of the total collective.

Patient Characteristics	*n* (%)
Age (years)	Median 58.0
	Range 69
Tumor foci	Unifocal 173 (54.0)
	Multifocal 147 (45.9)
Histology	NST 144 (49.8)
	Non-NST 154 (50.2)
Tumor grade	G1 or G2 107 (70.9)
	G3 44 (29.1)
pT	pT1 193 (66.8)
	pT2–pT4 96 (33.2)
pN	pN0 165 (57.5)
	pN1–pN3 122 (42.5)

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
