# Peer review of "Hormone Receptor Expression in Multicentric/Multifocal versus Unifocal Breast Cancer: Especially the VDR Determines the Outcome Related to Focality"

_ijms, 2019, doi:10.3390/ijms20225740_

Round 1

Reviewer 1 Report

I have reviewed and agree with the changes.

Reviewer 2 Report

No further comments

This manuscript is a resubmission of an earlier submission. The following is a list of the peer review reports and author responses from that submission.

Round 1

Reviewer 1 Report

lines 51 and 52: I agree that the endocrine therapy is indispensable and reduce the mortality and risk of relapse regardless of age. But NCCN does not usually recommend endocrine therapy in T1a N0 patients. So "regardless of TNM stage " may not be true.

The manuscript is novel and agree with author`s discussion. However, the manuscript may benefit from editing by an expert in English grammar. Some of the sentences are too long and some of the sentences can be changed to make it easy for readers. example: Lines 59,60 and 61: " a high VDR expression in BC.........decision making".

lines 226,227: "in contrast .....when conducting MV analysis"

These are a few examples.

Author Response

The Following is a point-to-point response to the reviewer`s comments

Thank you for allowing us to revise our paper. The constructive advice of the peer reviewers has substantially improved our paper. Attached are our detailed responses to your comments.

Editor: lines 51 and 52: I agree that the endocrine therapy is indispensable and reduce the mortality and risk of relapse regardless of age. But NCCN does not usually recommend endocrine therapy in T1a N0 patients. So "regardless of TNM stage " may not be true.

Authors: Endocrine therapy regimes have decreased the BC associated mortality rate of approximately 30% and the risk for relapse of 40% significantly regardless of the age thus nowadays being according to NCCN guidelines an indispensable part of the adjuvant therapy for women with HR+ BC [2, 3].

Editor: The manuscript is novel and agree with author`s discussion. However, the manuscript may benefit from editing by an expert in English grammar. Some of the sentences are too long and some of the sentences can be changed to make it easy for readers.

Authors: Thanks for the reviewer’s reminding, the revision manuscript has been proofread by a native English speaker.

Lines 59,60 and 61: " a high VDR expression in BC.........decision making".

Authors: In BC VDR expression is inversely associated with a higher cancer incidence, disease progression and worse prognosis [5, 13]. Anyway, only the ER, PR and HER2 status are currently considered as important predictive and prognostic immunohistochemical markers in the therapy decision making [4, 5, 14].

lines 226,227: "in contrast .....when conducting MV analysis"

Authors: In contrast, our study evaluated by multivariate analysis that the presence of PR provided no independent predictive value for the OS.

Reviewer 2 Report

This manuscript analyzed the association of hormone receptors and patient survivals by multifocal versus unifocal breast cancers. The authors found ER, PR expressions are good prognosis factors for BC patients, whereas TNBC and VDR expressions are strong associated with poor parent survival rate. In general, most of the finding has been well known. Therefore, the authors should find some novelty to the further investigation. The review command is listed below.

1: The presentation of result section is not well-organized. The assessment between each receptor should be equally displayed. For example: ER and PR showed only in unifocal BC, whereas VDR showed only in multifocal BC. It is hard to identify the main issue and new finding through the text.

2: The detail information in tables should be given, such as patient number, survival years and etc. It is inappropriate to show only significances.

3: It is relatively hard to read the whole manuscript. This manuscript should be English edited by professionals.

4: The issue of hormone receptor expressions in multicentric/multifocal versus unifocal breast cancers has been well-established by many investigators. Therefore, some novelty is requires in this manuscript. The relate studies are listed below.

World Journal of Surgical Oncology 2014, 12:266

BMC Cancer volume 13, Article number: 361 (2013)

Middle East Journal of Cancer; April 2016; 7(2): 69-78

Rom J Morphol Embryol 2019, 60(1):103–110

Romanian Society of Ultrasonography in Obstetrics and Gynecology [13] 59-64 [2017]

Pathology Research International Volume 2011, Article ID 480960, 5 pages

5: The VDR expression in breast cancer is also not well analyzed in this study. The VDR expression in scores (1+ to 4+) should be presented and further associated with patients survival. The molecular oncogenic mechanism of VDR in BC is also suggested to be discussed.

6: It is also not clear that the meaning on numbers (months or rears) in patient survivals of OS and DFS. The definition is not given in either figure legend or materials and methods section.

Author Response

The Following is a point-to-point response to the reviewer`s comments

Thank you for allowing us to revise our paper. The constructive advice of the peer reviewers has substantially improved our paper. Attached are our detailed responses to your comments.

This manuscript analyzed the association of hormone receptors and patient survivals by multifocal versus unifocal breast cancers. The authors found ER, PR expressions are good prognosis factors for BC patients, whereas TNBC and VDR expressions are strong associated with poor parent survival rate. In general, most of the finding has been well known. Therefore, the authors should find some novelty to the further investigation. The review command is listed below.

1: The presentation of result section is not well-organized. The assessment between each receptor should be equally displayed. For example: ER and PR showed only in unifocal BC, whereas VDR showed only in multifocal BC. It is hard to identify the main issue and new finding through the text.

Authors: We described each receptor in its own subheading (2.1-2.4), while graphically summarizing the new findings at the end of each paragraph e.g.

Table 6. Significant results for the vitamin receptor positive patients.

VDR

Unifocal

Multifocal

Overall survival

Disease- free survival

· -

Grading

· +

pT

·+

pN

·+

pM

· -

= Expression of the particular receptor has a significant influence of the marked characteristics;

+ = Receptor expression effects the marked characteristics significant positively;

- = Receptor expression effects the marked characteristics significant negatively.

2: The detail information in tables should be given, such as patient number, survival years and etc. It is inappropriate to show only significances.

Authors: We have edited all table and figure legends. Please find below the table with the detailed patient characteristics we inserted in the Material and Method section.

Table 9. Patient Characteristics of the total collective: sample size of receptor expression.

Receptor expression

Unifocal

Multifocal

ER+

105

96

ER-

28

17

PR+

79

75

PR-

54

38

VDR+

66

60

VDR-

89

57

TN

15

5

3: It is relatively hard to read the whole manuscript. This manuscript should be English edited by professionals.

Authors: Thanks for the reviewer’s reminding, the revision manuscript has been proofread by a native English speaker.

4: The issue of hormone receptor expressions in multicentric/multifocal versus unifocal breast cancers has been well-established by many investigators. Therefore, some novelty is requires in this manuscript. The relate studies are listed below.

World Journal of Surgical Oncology 2014, 12:266

BMC Cancer volume 13, Article number: 361 (2013)

Middle East Journal of Cancer; April 2016; 7(2): 69-78

Rom J Morphol Embryol 2019, 60(1):103–110

Romanian Society of Ultrasonography in Obstetrics and Gynecology [13] 59-64 [2017]

Pathology Research International Volume 2011, Article ID 480960, 5 pages

Authors: Generally, unifocal BC is associated with a better prognosis, including OS and DFS than multifocal and/or multicentric BC with identical tumor size. Weissenbacher et al. [25] even declare the focality as an independent prognostic factor and hypothesize that it should be considered in the current TNM classification of the UICC. In a following study they demonstrated that the down regulation of E-cadherin multicentric/multifocal compared to unifocal BC with identical TNM-staging might be connected with the worse prognosis of this tumor type [26]. In contrast Fushimi et. al. [27] did not find the focality to be predictive of a worse prognosis in their study of 734 BC patients since it did not sicnificantly influence the OS. Nevertheless, they confirmed that multifocal and multicentric BC is associated with a worse DSF (p=0.004). Another retrospective study, including 507 patients, claimed similar DSF and OS rates for each focality type and the lymph node status to be the only statistically significant factor affecting the prognosis[28]. In 2010 Tot et. al. analyzed in 875 cases the HR expression (ER, PR, HER2) in multifocal and unifocal BC, where they could not verify significant differences between these two tumortypes, regarding HR expression [29].

5: The VDR expression in breast cancer is also not well analyzed in this study. The VDR expression in scores (1+ to 4+) should be presented and further associated with patient’s survival. The molecular oncogenic mechanism of VDR in BC is also suggested to be discussed.

Authors: The VDR is expressed in healthy as well as in cancer cells. Activated by its hormonal ligand Calcitriol (1,25-Dihydroxyvitmin-D), it exerts its actions in an endocrine, paracrine and autocrine manner. When Calcitriol binds to the VDR, it causes a dimerization with the RXR (retinoid X receptor). The Calcitriol-VDR-RXR complex binds to VDREs (vitamin D response elements) in various regulatory regions, which induce transcriptional regulation of gene expression. These target genes intervene in molecular pathways, resulting in the diverse vitamin-D-mediated anticancer effects: inhibiting proliferation, inflammation, angiogenesis, invasion, metastasis and on the other side stimulating apoptosis and differentiation [9]. In this context, especially the two calcitriol-metabolizing catabolic enzymes CYP24A1 and CYP27B1 in cancer tissue gained importance for the carcinogenic effects of the pathway [10, 11]. Compelling evidence implies that the usual balance between CYP24A1 and CYP27B1 becomes dysregulated during carcinogenesis, leading to abrogation the tumor suppressive effects triggered by VDR [8].

In mammary gland the VDR expression fluctuates during the evolvement of a female body, inducting during puberty and peaking during pregnancy and lactation [8, 12]. In BC VDR expression is inversely associated with a higher cancer incidence, disease progression and worse prognosis [5, 13]. Anyway, only the ER, PR and HER2 status are currently considered as important predictive and prognostic immunohistochemical markers in the therapy decision making [4, 5, 14]. For the prophylaxis of the side effects of cancer therapy a supplementation of vitamin-d with a dose between 800 – 100 IU/d is common [15]. Already in 2009 Goodwin et. al showed for the first time, that there is a significant relation between low vitamin-d (25-Hydroxyvitamin D) serum levels and a higher risk for relapse and mortality for BC patients [16-18]. Also, compelling evidence was provided for the tumor suppressor function of VDR and agonists in breast tissue [9, 19].

More detailed analysis of the intensity and distribution of VDR receptor expression according to the IRS score (1-4), revealed that only a VDR expression equal or higher than an IRS of 3 influences the DFS significantly (p=0,024). (VDR1 p=0,070, VDR2 p=0,070)

6: It is also not clear that the meaning on numbers (months or rears) in patient survivals of OS and DFS. The definition is not given in either figure legend or materials and methods section.

Authors: We appreciate the reviewer’s recommendation, we have edited all figure and table legends in our manuscript.

Reviewer 3 Report

The sample size in each group should be added in tables, figures or figure legends. For example, how many patients with ER positive and negative was showed in Kaplan-Meier survival analysis (Figure 1)? In discussion, the authors described that “While some authors found by multivariate analysis the focality to be an independent prognostic factor, others claim that only the sum of the invasive diameters of multifocal BC could have an important impact on the prognosis. Did the authors try to evaluate the prognostic impact hormone receptor expressions via multivariate analysis, according to expression of ER, PR, and VDR on the multifocal and unifocal breast cancer?

Author Response

The Following is a point-to-point response to the reviewer`s comments

Thank you for allowing us to revise our paper. The constructive advice of the peer reviewers has substantially improved our paper. Attached are our detailed responses to your comments.

The sample size in each group should be added in tables, figures or figure legends. For example, how many patients with ER positive and negative was showed in Kaplan-Meier survival analysis (Figure 1)?

Authors: Thanks for the reviewer’s suggestion. Please find below the table with the detailed patients characteristics we inserted in the material and method section of the revised manuscript.

Table 9. Patient Characteristics of the total collective.

Receptor expression

Unifocal

Multifocal

ER+

105

96

ER-

28

17

PR+

79

75

PR-

54

38

VDR+

66

60

VDR-

89

57

TN

15

5

In discussion, the authors described that “While some authors found by multivariate analysis the focality to be an independent prognostic factor, others claim that only the sum of the invasive diameters of multifocal BC could have an important impact on the prognosis. Did the authors try to evaluate the prognostic impact hormone receptor expressions via multivariate analysis, according to expression of ER, PR, and VDR on the multifocal and unifocal breast cancer? 

Authors: Yes, please find the outcome of the multivariate analysis in the result section of our manuscript. The significant findings are displayed in tables in the result section. E.g.

Table 8. Multivariate Cox regression analysis of multifocal BC patients regarding OS.

Variable

Coefficient

HR (95%CI)

P Value

Age

0,050

1,052 (1,022-1,083)

0,001

Grading

-0,001

0,999 (0,992-1,007)

0,864

pT

0,642

1,900 (1,394-2,589)

0,000

pN

0,011

1,011 (1,000-1,022)

0,043

pM

2,390

10,914 (4,456-26,731)

0,000

VDR

-0,218

0,804 (0,671-0,964)

0,019

Significant results are shown in bold; HR: hazard ratio; CI: confidence interval.

Round 2

Reviewer 2 Report

This revised manuscript seems to not fulfill the requests that gave last time. Several important issues are not responded through this revision.

1: The Kaplan-Meier survival analysis in VDR negative and positive patients with Unifocal is still missing in this revised manuscript.

2: The characteristics of all breast cancer patients is far more too simple. A detail information should be given. Please see take the following paper as an example.

Prediction of Breast Cancer Survival Using Clinical and Genetic Markers by Tumor Subtypes. PLoS One. 2015; 10(4): e0122413.

3. The scoring analysis is still missing. This is a very critical analysis for both pathology and investigator. Without this data, the readers are not able to understand how the authors to define strong or weak VDR expression of breast cancers, and even believe the association in this manuscript.